# Load Recognition in Home Energy Management Systems Based on Neighborhood Components Analysis and Regularized Extreme Learning Machine

**DOI:** 10.3390/s24072274

**Published:** 2024-04-02

**Authors:** Thales W. Cabral, Fernando B. Neto, Eduardo R. de Lima, Gustavo Fraidenraich, Luís G. P. Meloni

**Affiliations:** 1Department of Communications, School of Electrical and Computer Engineering, University of Campinas, Campinas 13083-852, Brazil; t264377@dac.unicamp.br (T.W.C.); gfraiden@unicamp.br (G.F.); 2Companhia Paranaense de Energia, Curitiba 81200-240, Brazil; fernando.bauer@copel.com; 3Department of Hardware Design, Instituto de Pesquisa Eldorado, Campinas 13083-898, Brazil; eduardo.lima@eldorado.org.br

**Keywords:** machine learning, household appliances, active power, appliance recognition

## Abstract

Efficient energy management in residential environments is a constant challenge, in which Home Energy Management Systems (HEMS) play an essential role in optimizing consumption. Load recognition allows the identification of active appliances, providing robustness to the HEMS. The precise identification of household appliances is an area not completely explored. Gaps like improving classification performance through techniques dedicated to separability between classes and models that achieve enhanced reliability remain open. This work improves several aspects of load recognition in HEMS applications. In this research, we adopt Neighborhood Component Analysis (NCA) to extract relevant characteristics from the data, seeking the separability between classes. We also employ the Regularized Extreme Learning Machine (RELM) to identify household appliances. This pioneering approach achieves performance improvements, presenting higher accuracy and weighted F1-Score values—97.24% and 97.14%, respectively—surpassing state-of-the-art methods and enhanced reliability according to the Kappa index, i.e., 0.9388, outperforming competing classifiers. Such evidence highlights the promising potential of Machine Learning (ML) techniques, specifically NCA and RELM, to contribute to load recognition and energy management in residential environments.

## 1. Introduction

The rising demand for electrical energy presents a challenge to sustainable consumption, affecting various sectors. According to Kim et al. [1], considering diverse sources like biomass and natural gas, the residential sector contributes 27% to global consumption. Moreover, as per Rashid et al. [2] and Bang et al. [3], due to malfunctioning appliances and improper consumption habits, 30% of energy is wasted. One way to contribute to sustainable consumption and minimize such issues is to adopt Home Energy Management Systems (HEMS).

HEMS refers to technologies developed to manage the electricity consumption in households or commercial buildings. According to Motta et al. [4], a HEMS architecture consists of a controller and smart outlets to connect household appliances to the electrical grid. The electricity demand of a household may exhibit seasonal variations, influenced by the type of appliances in operation, such as heaters and air conditioning devices. However, HEMS is capable of monitoring such appliance activities, incorporating additional functionalities such as load disaggregation techniques presented by Lemes et al. [5], methods for anomaly detection in appliances as per Lemes et al. [6], and load recognition mechanisms as reported by Cabral et al. [7].

Load recognition means identifying appliances in operation and emerges as an essential building block for advancing home energy management systems. Load recognition also contributes to load disaggregation methods, allowing specific identification of devices after the disaggregation procedure. Furthermore, load recognition is fundamental to building robust appliance databases by analyzing electrical signals, making this process more precise and automatic. In the diversified domestic environment, where various appliances such as microwaves, dishwashers, air conditioning, freezers, and heaters operate, HEMS’ capability to determine which appliance is operating is indispensable. This functionality takes on a relevant practical dimension when replacing appliances connected to the smart outlets. In this case, the HEMS automatically identifies the new appliance in operation with the load recognition system. The same practical importance can be observed for automated database construction.

Currently, there is a trend towards using Machine Learning (ML) in state-of-the-art solutions for load recognition. Some works employ models considered computationally costly, such as Mian Qaisar and Alsharif [8] with the Artificial Neural Network (ANN), De Baets et al. [9], Faustine and Pereira [10], and Matindife et al. [11] with the Convolutional Neural Network (CNN), and Huang et al. [12] and Heo et al. [13] with the Long Short-Term Memory (LSTM). Other studies apply architectures with lower computational costs, such as Qaisar and Alsharif [14] with k-Nearest Neighbors (*k*-NN), Soe and Belleudy [15] with Classification and Regression Trees (CART), Qaisar and Alsharif [14] and Soe and Belleudy [15] with Support Vector Machine (SVM), and Zhiren et al. [16] with the standard Extreme Learning Machine (ELM). Furthermore, reducing the amount of information needed to identify the appliance in operation is a pertinent feature of the solutions. According to Cabral et al. [7] and Soe and Belleudy [15], it is feasible to achieve this using ML techniques for feature extraction, such as Linear Discriminant Analysis (LDA), Principal Component Analysis (PCA), and others. Conversely, we can limit the amount of information required regarding household appliances by solely using a single type of electrical signal, such as voltage, current, reactive power, or active power.

Designing an approach that guarantees high performance, reliability, and the  short training time of ML models using only a single type of electrical signal represents a crucial challenge in the load recognition area. Several approaches are seeking to achieve these objectives. Presently, the modern methods for load recognition use images generated from electrical signals such as voltage, current, active power, and others. The advantages of this approach are substantiated by the works as Faustine and Pereira [10], Matindife et al. [11], Gao et al. [17], De Baets et al. [9], and Cabral et al. [7]. In this regard, the motivation of this study is to propose enhancements to the load recognition system in HEMS. To achieve this, the current study utilizes images generated exclusively from the active power data of household appliances. The present work contributes to advancements in the load recognition area, introducing novel applications not previously explored in the literature. Our analysis investigates the underlying factors, highlighting Neighborhood Component Analysis (NCA) as a promising alternative technique for feature extraction in load recognition. Besides reporting the quantitative improvements over existing methods, we also emphasize the qualitative benefits, such as the enhanced system reliability through employing the Regularized Extreme Learning Machine (RELM) classifier instead of the standard ELM. Furthermore, our work seeks to optimize the capabilities of ML models to the maximum through the combination of Grid Search (GS) with K-fold Cross-Validation (K-CV). The results of the proposed system reveal the highest accuracy values, weighted average F1-Score (F_1_), and Kappa index (κ) when compared to the most modern methods in the literature. The innovations implemented also guarantee low training time. These results confirm the superiority of the innovations proposed in this manuscript. Furthermore, our solution is part of the research project under development named Open Middleware and Energy Management System for the Home of the Future. The project is a collaboration between the University of Campinas, the Eldorado Research Institute, and the Brazilian energy supplier Companhia Paranaense de Energia (COPEL).

### Major Contributions

The principal contributions of our work are as follows:Gaps like enhancing classification performance via approaches dedicated to separability between classes and models that reach improved reliability remain open. According to Manit and Youngkong [18], NCA provides enhanced class separability. In this study, we adopt NCA to extract relevant characteristics from the data, seeking the separability between classes to improve classification performance. We also apply the RELM to achieve higher reliability when identifying household appliances. It is relevant to mention that this work is the first to use NCA and RELM for load recognition. Besides the RELM-NCA pair, it is the first to verify the NCA-ELM pair for this task;The values obtained for the performance metrics reveal the promising potential of ML techniques, specifically NCA and RELM, to contribute to load recognition. Using the ‘Personalised Retrofit Decision Support Tools for UK Homes using Smart Home Technology’ (REFIT), the proposed approach achieved 97.24% accuracy and 97.14% F1. As well as via the Reference Energy Disaggregation Dataset’ (REDD), the method reached 96.53% accuracy and 96.48% F1. For both, the proposed approach has achieved performance improvements in load recognition. For the REFIT dataset, the difference between RELM and ELM reveals a 0.36% accuracy advantage for the RELM model. This advantage is the same for RELM compared to the SVM of the state-of-the-art system proposed in Cabral et al. [7]. For the REDD dataset, RELM’s advantage over SVM is 0.21%. RELM’s advantage over ELM for the same dataset is 2.71% accuracy. Furthermore, we can see a trend favoring the proposed method when we examine the accuracy of other state-of-the-art methods from the references. When comparing the best result of the proposed method, 97.24% accuracy, with the third-placed method in Qaisar and Alsharif [14], the difference is 1.84%;Our method provides an ultra-low training time of 0.082 s with the REFIT database, less than the SVM of the technique reported in Cabral et. al [7], which has a time of 0.469 s. This result means that the proposed approach is approximately 5.72 times faster than the competitor, representing a time saving of 82.52% compared to the competitor. Concerning ELM, the proposed approach is 2.33 times faster and saves approximately 57.07% of the time. When checking the REDD database, the proposed method has a training time of 0.123 s, while the SVM has 0.167 s. In this case, the proposed method is 1.36 times faster than the SVM, saving approximately 26.35% of the time. For the REDD dataset, the proposed approach requires more time than ELM to complete its training process. However, only the proposed approach has the shortest achievable time compared to the other methods: 0.082 s.

The structure of the remainder of this paper is as follows: Section 2 provides a detailed background to lay the foundation for this study. Section 3 provides a detailed description of the proposed system, elucidating the implemented processing flow. This section includes detailed input data, the feature extraction technique employed, the criteria for selecting appropriate components, and the implementation of machine learning models. Section 4 presents the metrics utilized in this manuscript, accompanied by a justification for their choice. Section 5 examines the results obtained from the proposed system when employing two databases. Furthermore, this section discusses the outcomes, offering insights and interpretations of the findings. Section 6 concludes the paper, summarizing the contributions, implications of the proposed strategy, and potential future work.

## 2. Background

This section presents the related works in the literature and introduces the theoretical principles, such as the feature extraction technique and the architecture of the ML model.

### 2.1. Related Works

In the literature, there are a variety of strategies that perform load recognition. In Qaisar and Alsharif [14], the authors use active power. Active power performs work in an electrical system and refers to the energy consumed by the appliance to operate. However, the study uses different procedures to classify each device category. The paper uses devices from the Appliance Consumption Signature-Fribourg 2 (ACS-F2) database and the accuracy metric to evaluate the performance of the SVM and kNN models at the devices’ classification stage. In Cabral et al. [7], the authors exclusively use the active power profile of the household appliances in the REDD and REFIT datasets. In this reference, the *k*-NN, Decision Tree (DT), Random Forest (RF), and SVM models can perform load recognition with a balance between short training time and high performance for the accuracy metrics, weighted average F1 score, and Kappa index.

Nonetheless, approaches do not necessarily only exploit the active power of a household appliance. The study reported by Mian Qaisar and Alsharif [8] uses active and reactive power. Reactive power in an alternating current system does not perform effective work, being stored and returned to the electrical system due to the presence of reactive elements, such as capacitors and inductors. In this case, the authors also apply accuracy to evaluate the performance of the ANN and *k*-NN models. In Matindife et al. [11], the researchers use a private dataset involving active power, current, and power factor. Here, by employing the Gramian Angular Difference Field (GADF) for feature extraction; the researchers convert the data into images. In the sequence, the CNN recognizes the appliances and the authors test the robustness of their proposal using the recall, precision, and accuracy metrics.

On the other hand, alternative studies utilize different types of data, as demonstrated in Borin et al. [19], which employs instant current measurements. The study applies Vector Projection Classification (VPC) in pattern recognition of loads. In this reference, the authors assess the performance of the proposed approach through the percentage of identified devices. Some methods utilize a combination of these other variables, such as voltage and current, for instance. In Zhiren et al. [16], the study uses a private dataset. The authors evaluate the proposed solution through accuracy metric, where the models tested are ELM, Adaboost-ELM, and SVM. In Faustine and Pereira [10], the scientists employ the Plug Load Appliance Identification Dataset (PLAID) dataset and the F_1_ example-based (F_1_-eb) and F_1_ macro-average (F_1_-macro) metrics in their analysis. The methodology proposed in this reference focuses on the Fryze power theory, which decomposes the current characteristics into components. As a result, the current becomes an image-like representation and a CNN recognizes the loads.

Unlike the utilization of voltage and current profiles, certain studies consider alternative attributes. Heo et al. [13] use Amplitude–Phase–Frequency (APF). The researchers employ accuracy and the F_1_-Score as metrics to evaluate the overall performance of the proposed system and the following databases: Building-Level fully labeled Electricity Disaggregation (BLUED), PLAID, and a private database. As reported in the study, the use of HT-LSTM improves the recognition of devices with differences in the transient time and transient form of the load signal. Furthermore, the proposed scheme includes an event detection stage. Event detection is not always present in the strategies published in the literature but it is a tool that allows the system to identify when the appliance has turned on and off. The references Cabral et al. [7], Anderson et al. [20], Norford and Leeb [21], and Le and Kim [22] contain event detection strategies. It is relevant to mention that event detection is not the focus of our proposed work. Nevertheless, we use Wavelet transform to detect the ON/OFF status of the appliances according to references Lemes et al. [6] and Cabral et al. [7]. The selection of the Wavelet transform is justified due to its ability to detect appliance activity simply through the analysis of the level 1 detail coefficient. According to Lemes et al. [6], level 1 already contains enough information to detect ON/OFF activity. Hence, detecting the ON/OFF activity of the appliance can be performed without needing to decompose the signal into higher levels.

Another significant factor is the volume of data involved in the proposed approaches; more attributes to consider mean a more computationally complex and invasive system. In Huang et al. [12], the authors consider the steady-state power, the amplitude of the fundamental wave of the transient current, the transient voltage, and the harmonics of the transient current. In this case, the work adopts the REDD dataset and F_1_-Score in the tests. The methodology combines LSTM layers with Back Propagation (BP) layers, resulting in the following architecture: the Long Short-Time Memory Back Propagation (LSTM-BP) network. The method described by Soe and Belleudy [15] uses characteristics from the active power of the equipment present in the Appliance Consumption Signature-Fribourg 1 (ACS-F1). Such features are the maximum power, average power, standard deviation, number of signal processing, operating states, and activity number of the appliances. The article evaluates the performance of the SVM, *k*-NN, CART, LDA, Logistic Regression (LR), and Naive Bayes (NB) models in terms of accuracy. It is worth emphasizing that as the diversity of electrical signals and parameters demanded from home appliances increases, the load recognition method becomes more intrusive and computationally expensive. For this reason, creating a strategy that has an optimal balance between high performance, reliability, and short training time based on a single type of electrical signal represents a key challenge in the load recognition field.

Finally, it is essential to mention some shortcomings in the methods proposed in the literature. Few studies use only one type of electrical signal in their approaches. The greater the number of electrical signals involved, the more invasive and computationally costly the method becomes. For example, the works of Mian Qaisar and Alsharif [8], Qaisar and Alsharif [14], and Zhiren et al. [16] use many parameters excessively. On the other hand, the majority of existing studies do not include a stage for detecting the ON/OFF status of the equipment, for example, the works of Mian Qaisar and Alsharif [8], Soe and Belleudy [15], and Zhiren et al. [16]. This condition limits the practical use of these methods. Most studies in the literature do not consider applying procedures to optimize their approaches, such as the hyperparameter search, for example, Faustine and Pereira [10], Qaisar and Alsharif [14], and Soe and Belleudy [15]. Adopting this procedure supports the definition of classifier structural parameters and can provide additional performance gains. Other papers are not concerned with evaluating the reliability of the system.

### 2.2. Feature Extraction

Feature extraction concerns the process of transforming relevant characteristics from raw data to create more compact and informative representations. The extracted features describe distinctive properties of the data and practitioners widely apply this approach across several areas, such as image processing, according to Nixon and Aguado [23] and Chowdhary and Acharjya [24], signal processing, in line with Gupta et al. [25] and Turhan-Sayan [26], and ML according to Musleh et al. [27] and Kumar and Martin [28]. One of the advantages of some feature extraction techniques is the reduction in data dimensionality, thereby decreasing computational complexity.

Several techniques exist for feature extraction. The approach choice depends on the nature of the data, the task concerned, and the computational cost involved. Some studies, according to Veeramsetty et al. [29] and Laakom et al. [30], employ autoencoders for compact data representations. However, this kind of architecture can make methods computationally expensive. Alternative investigations use computational techniques that are less resource-intensive, such as in Reddy et al. [31] with LDA, Fang et al. [32] with Independent Component Analysis (ICA), and Bharadiya [33] and Kabir et al. [34] with PCA.

Currently, more modern methods employ NCA to eliminate redundant information to reduce computational cost, according to Ma et al. [35]. NCA is a technique focusing on learning a distance metric in the feature space, optimizing the similarity between points without necessarily decreasing the dimensionality of the data. As per Goldberger et al. [36], the NCA technique is based on *k*-NN and stands out for optimizing a distance metric to enhance the quality of features, especially in classification tasks where the distinction between classes is crucial.

According to Singh-Miller et al. [37], the NCA algorithm uses the training set as the input, i.e., $\left\{x_{1},x_{2},\cdots ,x_{Q}\right\}$ with $x_{i}$∈$R^{Q}$, and $y_{i}$ with the set of labels $\left\{y_{1},y_{2},\cdots ,y_{Q}\right\}$. The algorithm needs to learn a projection matrix A of dimension q×Q, which it uses to project the training vectors $x_{i}$ into a low-dimensional representation of dimension *q*. To obtain low-dimensional projection, NCA requires learning a quadratic distance metric γ, that optimize the performance of *k*-NN. The distance γ between two points, $x_{i}$ and $x_{j}$, is
(1)$$\gamma_{ij}=\gamma \left(x_{i},x_{j}\right)=\;{\left\|A\left(x_{i}-x_{j}\right)\right\|}^{2},$$
where ‖·‖ is the Frobenius norm and A is a linear transformation matrix. In the NCA technique, each point *i* chooses another point *j* as its neighbor from among *k* points with a probability $P_{ij}$ and assumes the class label of the selected point. According to Goldberger et al. [36], through $\gamma_{ij}$, $P_{ij}$, and the optimization of the objective function f(A), NCA calculates a vector representation in a low-dimensional space ($x_{i}^{\left(q\right)}$). The vector in low-dimensionality can be represented by
(2)$$x_{i}^{\left(q\right)}=Ax_{i},$$

It is possible to produce a matrix in low-dimensional space, depending on the implementation. This scenario is subject to the input data and the number of components required for the application. Detailed information regarding the NCA algorithm is available at Goldberger et al. [36].

### 2.3. Extreme Learning Machine (ELM)

The ELM presents a visionary structure in the ML field, standing out for its computational efficiency and conceptual simplicity. In contrast to many conventional neural networks, where all parameters need adjustment during training, ELMs adopt a unique strategy by randomly fixing the weights of the hidden layer and focusing solely on learning the weights of the output layer. This methodology enables a short training time. Furthermore, the simplified architecture of ELMs facilitates implementation, making them an appealing choice for applications requiring computational efficiency and robust performance in supervised learning tasks.

As per the formal description of ELM in accordance with Huang et al. [38], for different samples $\left(x_{i},t_{i}\right)$, where $x_{i}={\left[x_{i1},x_{i2},\cdots ,x_{in}\right]}^{T}\in R^{n}$ and $t_{i}={\left[t_{i1},t_{i2},\cdots ,t_{im}\right]}^{T}\in R^{m}$, the output of an ELM is
(3)$$\sum_{i=1}^{L}\beta_{i}g\left(\langle w_{i}\;,\;x_{j}\rangle +b_{i}\right)=t_{j},\;\;j=1,2,3,\;\cdots ,N,$$
in which $w_{i}={\left[w_{i1},w_{i2},\cdots ,w_{in}\right]}^{T}$ is the input weight vector connecting the *i*th hidden neuron, $\beta_{i}={\left[\beta_{i1},\beta_{i2},\cdots ,\beta_{im}\right]}^{T}$ is the output weight vector connecting the *i*th hidden neuron, $b_{i}$ is the bias, *L* is the number of hidden neurons, $g\left(\cdot \right)$ is the activation function, and 〈·,·〉 is the inner product. Nevertheless, the Equation (Equation 3) can be written in the matrix form as
(4)Hβ=T

H is the output matrix of the hidden layer of the neural network and can be expressed as follows:   
(5)$$H=\left[\begin{array}{ccc} g\left(\langle w_{1}\;,\;x_{1}\rangle +b_{1}\right) & \cdots & g\left(\langle w_{L}\;,\;x_{1}\rangle +b_{L}\right)\\ \vdots & \ddots & \vdots \\ g\left(\langle w_{1}\;,\;x_{N}\rangle +b_{1}\right) & \cdots & g\left(\langle w_{L}\;,\;x_{N}\rangle +b_{L}\right) \end{array}\right],$$
where
(6)$$\beta =\left[\begin{array}{c} \beta_{1}^{T}\\ \vdots \\ \beta_{L}^{T} \end{array}\right]\;\mathrm{and}\;\;T=\left[\begin{array}{c} t_{1}^{T}\\ \vdots \\ t_{N}^{T} \end{array}\right]$$

However, we can solve the system described in (Equation 4) through the Moore–Penrose pseudo-inverse of the H, depicted as $H^{†}$, where $H^{†}=\left[{\left(H^{T}H\right)}^{-1}H^{T}\right]$. Consequently, we can determine the output weights by
(7)$$\hat{\beta }=H^{†}T=\left[{\left(H^{T}H\right)}^{-1}H^{T}\right]T$$

Figure 1 illustrates the standard ELM, one of the ML models of the proposed system. Later in the manuscript, our approach includes modifications to the standard ELM to achieve enhanced results.

## 3. Proposed System

Figure 2 provides an overview of the proposed system for load recognition, where the collection and transmission of data are carried out through smart outlets and the controller, respectively. Both the smart outlets and the controller are illustrated in the blue color. In the second panel, the collected data are highlighted in blue, while the preliminary data processing blocks are represented in light gray. In the sequel, we have the feature extraction stage utilizing the NCA technique, represented in red. Subsequently, in dark gray, in the last part of the flowchart, Figure 2 depicts the stage responsible for the ML model optimization, aiming for the enhanced performance of the classifiers. At the end, depicted in light blue, the system presents the type of appliance in operation.

Figure 2a depicts the HEMS system consisting of the controller and smart outlets. The controller enables the processing of data either locally or its transmission to a cloud server. Moreover, it can execute pre-trained algorithms and send consumption alerts to the end user. The solution is minimally invasive. In practice, users only need to provide internet access to the controller and pair it with the smart outlets.

According to Figure 2b, the system features an ON/OFF operational state detection stage via Discrete Wavelet Transform (DWT) to determine when the appliance is in operation. An appliance registers as operating when it exhibits non-zero active power values, even after employing a prior filtering of potential noise. To determine whether an appliance is in operation as demonstrated in Lemes et al. [6], we only need the level-1 detail coefficients obtained by decomposing the active power using DWT with the Daubechies 4 mother wavelet. Subsequently, the system transforms the segments with the detected activities into images, as described in Cabral et al. [7]. In this procedure, the proposed strategy converts the curve representing the electrical activity of the appliance into black pixels and the background into white pixels, where the system has the flexibility to adjust the pixel resolution based on computational cost requirements. Therefore, each generated image contains a cycle of appliance activity. Following this, the system rearranges the image rows into a column vector of size *I*. Afterward, the method generates a set of vectors containing *N* images and places this set into a matrix S, with dimensions *I* × *N*.

Figure 2c illustrates the processing chain for feature extraction via NCA. In our case, we apply NCA to the pixel matrix S, transforming the data from S, with high dimensionality, into the matrix $S_{\left(\mathrm{NCA}\right)}^{\left(q\right)}$, with low dimensionality (q). Once we have the learned transformations applied to the data, we can estimate the variance of the transformed data for each dimension or component. Thus, it is feasible to assess the fraction of variance contained in each dimension and calculate the Cumulative Explained Variance (CEV) as more dimensions or components are incorporated. Then, we can choose the optimal number of components using CEV. The different components are defined based on the CEV, as per Algorithm 1. To detail this stage, Algorithm 1 specifies all the procedures of feature extraction through NCA. In particular, Algorithm 1 encompasses the sequence of procedures necessary to calculate the CEV, determine the number of components through CEV, use the NCA technique with the optimized number of elements, and obtain the transformed data via NCA.

Figure 2d represents the processing chain dedicated to optimizing the ML model. During this phase, the system feeds the ML models with the transformed data. For the ELM model, the processing chain conducts a hyperparameter search through GS and K-CV, employing the candidate set for the number of neurons and the number of folds. Algorithm 2 details the instructions for the ELM model optimization process. It is relevant to mention that our system can employ the optimization process in ELM or RELM. Nonetheless, there is a difference between the ELM and RELM models. Such a particularity is the regularization coefficient. This element is a parameter used in ML techniques, such as ridge regression, to control the model’s fit to the data. The function of the regularization coefficient is to impose a penalty on the magnitude of the model coefficients, preventing them from reaching excessively high values. Such a strategy minimizes the risk of overfitting by preventing the model from overly adjusting to the data.

The regularization coefficient is a mechanism that aids in achieving an appropriate balance between bias and variance. An elevated regularization coefficient leads to an increase in the bias and a decrease in the variance. In contrast, a small regularization coefficient allows for greater flexibility in model fitting to the data, promoting an increase in variance. Such a coefficient acts as a regulator, influencing the synthesis of a simpler or a more complex model. Therefore, appropriately adjusting the regularization coefficient results in a more balanced trade-off between bias and variance.

For this reason, the regularization coefficient inclusion prevents the model from becoming excessively specific to the training data, making it more robust to new datasets unseen during training. This effect enhances the capability of the model generalization. Thus, a model with a suitable regularization coefficient tends to preserve the relevance of identified patterns, even in different contexts from the training data. Therefore, it is necessary to carefully seek the appropriate value for the regularization coefficient, ensuring that the model effectively enhances its generalization capability.

**Algorithm 1** Feature extraction using neighborhood component analysis with component selection based on cumulative explained variance**Input:** Image dataset generated (S), initial number of components (ϕ), threshold (ψ)**Output:** training set S(tr,NCA)(q) e testing set S(ts,NCA)(q)
 1:first method:Split the S database and derive the S(tr) training data and the S(ts) test data. 2:second method:Train the NCA with S(tr), using ϕ initial components. After training, obtain the transformed dataset S(tr,NCA)(ϕ) 3:third method:Estimate the covariance matrix C(NCA), according to Lemes et al. [5], for the transformed data S(tr,NCA)(ϕ) 4:fourth method:Calculate the eigenvalues ei through C(NCA)=E·diag(e1,e2,⋯,eϕ)·E−1, where E is the eigenvectors matrix e diag(e1,e2,⋯,eϕ) is the diagonal matrix containing the eigenvalues 5:fifth method:Order the eigenvalues in descending sequence: e1≥e2≥e3≥⋯≥eϕ 6:sixth method:Compute the proportion of variance explained by each eigenvalue ρi=ei∑j=1ϕej 7:seventh method:Obtain the CEV for the *i*-th component: CEV_*i*_ = ∑j=1iρj 8:eighth method:Create the *q* variable to receive the optimized number of components and initialize *q* = 0Determine the optimized number of components   **if** CEV_*i*_ ≥ ψ      *q* ← number of *i*-th component,   **end if** 9:ninth method:Re-train NCA with S(tr), this time utilizing only *q* components. After to the training process, apply NCA to generate the transformed datasets for both the training set, denoted as S(tr,NCA)(q), and the test set, denoted as S(ts,NCA)(q)**return** S(tr,NCA)(q) e S(ts,NCA)(q)


Obtaining the most suitable regularization coefficient for the RELM model through trial and error can be a considerable challenge. Once again, this objective becomes feasible through hyperparameter search strategies, such as GS and K-CV. Structurally, adding a regularization coefficient to the ELM architecture to obtain the RELM model involves modifying the pseudo-inverse of **H**, as per Equation (Equation 8). This approach is an elegant strategy that ensures excellent results.
(8)$${\hat{\beta }}_{\lambda }=H^{†}T=\left[{\left(H^{T}H+\lambda I\right)}^{-1}H^{T}\right]T$$

Nevertheless, in RELM optimization, it is necessary to include candidates set for the regularization coefficient. In this phase, the system considers the set of candidates for the number of neurons, the set of candidates for the regularization coefficient, and the number of folds during the hyperparameter search. In this case, Algorithm 3 describes the RELM model optimization process in detail.

**Algorithm 2** Optimizing extreme learning machine through hyperparameter tuning with grid search and K-fold cross-validation for enhanced performance**Input:** Candidates for the number of neurons (η1,η2,⋯,ηn), S(tr,NCA)(q), S(ts,NCA)(q), number of folds (K)**Output:** Optimized ELM
 1:first method:Load the candidates for the hyperparameters: candidates for the number of neurons (η1,η2,⋯,ηn) 2:second method:Employ GS with K-CV   Divide S(tr,NCA)(q) in K folds   Train the model for each fold   Compute accuracy   Mean accuracy   Attributes the average accuracy to the present set of hyperparameters    Choose the hyperparameter set with the highest average accuracy achieved, i.e,    η(optimal) 3:third method:Train the model using η(optimal) 4:fourth method:Testing the optimized model with S(ts,NCA)(q)**return** Optimized ELM


**Algorithm 3** Optimizing regularized extreme learning machine through hyperparameter tuning with grid search and K-fold cross-validation for improved performance**Input:** Candidates for the number of neurons (η1,η2,⋯,ηn), candidates for the regularization coefficient (λ1,λ2,⋯,λn), S(tr,NCA)(q), S(ts,NCA)(q), number of folds (K)**Output:** Optimized RELM
 1:first method:Load the candidates for the hyperparameters: candidates for the number of neurons (η1,η2,⋯,ηn) and candidates for the regularization coefficient (λ1,λ2,⋯,λn) 2:second method:Employ GS with K-CV   Divide S(tr,NCA)(q) in K folds   Train the model for each fold   Compute accuracy   Mean accuracy   Attributes the average accuracy to the present set of hyperparameters   Choose the hyperparameter set with the highest average accuracy achieved, i.e,   λ(optimal) and η(optimal) 3:third method:Train the model using λ(optimal) and η(optimal) 4:fourth method:Testing the optimized model with S(ts,NCA)(q)**return** Optimized RELM


## 4. Performance Evaluation Metrics

To evaluate the performance of the proposed system, this work uses the metrics of accuracy, $F_{1}$, and κ, according to Cabral et al. [7]. These metrics rely on measures of true-positive (TP), true-negative (TN), false-positive (FP), and false-negative (FN).

The accuracy, defined in (Equation 9), is responsible for measuring the number of instances correctly classified in the test set and presents the overall performance of the models.
(9)$$\mathrm{Accuracy}=\frac{\mathrm{TP}+\mathrm{TN}}{\mathrm{TP}+\mathrm{FP}+\mathrm{TN}+\mathrm{FN}}$$

Depending on the nature of the devices, each appliance can produce a different number of events. It is possible to consider this effect in the F_1_-Score metric to provide a fair performance analysis. However, it is necessary to consider the size of the set of instances of a class (*d*) and the size of the dataset (*D*), as per Alswaidan and Menai [40] and Guo et al. [41]. To achieve it, this study employs the weighted average F_1_-score, denoted as F_1_ according to (Equation 10)
(10)$$\begin{array}{cc} F_{1} & =\frac{1}{D}\sum d\times F_{1}\text{-}\mathrm{Score}\\ \; & =\frac{1}{D}\sum d\times \left[\frac{2\times \mathrm{TP}}{2\times \mathrm{TP}+1\times (\mathrm{FN}+\mathrm{FP})}\right] \end{array}$$

This work applies the Kappa index to verify the agreement of the proposed strategy. As per Matindife et al. [11], Kappa operates within the interval of [−1,1]. A value of −1 signifies an absence of agreement, 0 represents agreement occurring by chance, and 1 indicates perfect agreement. This manuscript defines Kappa index, denoted as κ, by (Equation 11)
(11)$$\kappa =\frac{2\times \left(\mathrm{TP}\times \mathrm{TN}-\mathrm{FN}\times \mathrm{FP}\right)}{\left(\mathrm{TP}+\mathrm{FP}\right)\times \left(\mathrm{FP}+\mathrm{TN}\right)+\left(\mathrm{TP}+\mathrm{FN}\right)\times \left(\mathrm{FN}+\mathrm{TN}\right)}$$

Each metric offers a unique perspective on the performance of ML models, ensuring a holistic assessment. Accuracy is essential as it provides an overall success rate of the model. $F_{1}$ was selected to address a potential imbalance among classes in used datasets. This metric offers a more nuanced view of the model in scenarios when the class distribution is imbalanced. The κ evaluates the agreement between the predicted and observed classifications, providing insights into the model’s performance regarding reliability. By employing these three metrics together, we aim to present a comprehensive evaluation of the performance of our model, capturing its effectiveness in classification and its robustness in different evaluation parameters. This multifaceted approach allows us to validate the model’s utility in diverse scenarios, ensuring its reliability and applicability.

## 5. Results and Discussions

This study assesses the proposed solution using two databases, REDD from Kolter and Johnson [42] and REFIT from Murray et al. [43]. These databases encompass two households characterized by distinct measurement frequencies, types of equipment, and equipment number. The deliberate selection of these databases ensures a comprehensive evaluation of the method generalization. The initial procedure is to homogenize the resolution at 32×32 pixels for both databases. The system generates 4609 images from the REDD database and 2723 images from the REFIT database. Procedures involving training in the feature extraction and in the model optimization consider a data partition of 80% for training and 20% for testing in both databases, respectively. The proposed system applies feature extraction via NCA. Related work involving PCA indicates that the REFIT database requires more components than REDD. Concerning this, the proposed solution adopts ϕ = 300 for REFIT and ϕ = 100 for REDD. It is necessary to mention that these ϕ values are just initial values; our system endogenously determines the most appropriate number of components via CEV. For this task, we use a threshold of ψ = 0.99 for both REFIT and REDD. The threshold of ψ = 0.99 was selected based on previous studies that indicate the effective extraction of relevant components. For example, Cabral et al. [7] employ this value for the threshold to ensure that only components with significant contributions to the data variance are retained. In the sequence, the system initiates the optimization process for ML models. For both datasets, the hyperparameter tuning procedures use K = 10 folds. According to Kuhn et al. [44], opting for K = 10 is advocated because it can generate test error rate estimates unaffected by undue bias or excessive variance. Moreover, we also selected 10 to balance performance reliability and computational efficiency. In this search, the number of neurons ($\eta_{1},\eta_{2},\cdots ,\eta_{n}$) ranged from 100 to 1000, with a step size of 100, for the ELM and RELM models, respectively. The step size 100 was chosen to foster the model convergence without significantly compromising computational time. In addition, for the RELM model, we need to include the candidates for the regularization coefficient ($\lambda_{1},\lambda_{2},\cdots ,\lambda_{n}$) in the hyperparameter search. Thus, the set of the values for the regularization coefficient candidates ranged from 0.0001 to 0.1, with the search step increasing 10 times from one value to the next in the mentioned sequence.

### 5.1. Scenario Using the REFIT Dataset

For the first analysis scenario, this study utilizes the REFIT dataset. REFIT comprises active power measurements from 20 residences, recorded at a frequency of 1/8 Hz. For the REFIT scenario, we consider appliances of household 1. This household includes freezers, washer-dryers, washing machines, dishwashers, computers, televisions, and electric heaters. For this dataset, the system generated 4609 images representing the activities of the operational household appliances.

In this scenario, the system generates the results of Table 1 using the REFIT database from Murray et al. [43]. To preserve the layout of the manuscript, we show just some of the 300 components in Table 1. As highlighted in Table 1, using a threshold of ψ = 0.99 for the CEV, it is necessary to employ *q* = 228 components. Here, we use the GS and K-CV for a hyperparameter search of ELM and RELM. In this manner, for both models, the search specified the value of $\eta_{\left(\mathrm{optimal}\right)}$ = 400 for the number of neurons, respectively. Furthermore, the hyperparameter search determined $\lambda_{\left(\mathrm{optimal}\right)}$ = 0.01 for the RELM architecture.

Upon checking the comparison in Table 2, it is evident that a slight performance difference exists among the ELM and RELM models concerning the accuracy metric. However, the RELM architecture exhibited the highest value, 97.24%, for accuracy. This advantage persists for the RELM model in the F_1_ metric, with a value of 97.14%. Regarding the agreement between the predicted and expected values, the RELM model achieves the highest value of κ, 0.8300.

Table 3 compares the training times of the ELM and RELM models. In this scenario, the ELM classifier has the longest training time, at 0.191 s, followed by the RELM at 0.082 s. These results highlight the RELM model as having the shortest training time.

It is important to note that the obtained results suggest that the RELM architecture outperforms the previous model, ELM. We can extend this analysis further by comparing the second-best model in the literature for the same task, SVM in Cabral et al. [7], which achieves 96.88% accuracy, 96.61% to F_1_, and 0.8375 to κ. In contrast, the RELM classifier surpasses these metrics with values of 97.24% accuracy, 97.14% to F_1_, and 0.8300 to κ. Additionally, when examining the training time among the mentioned models, SVM has a longer training time in seconds, 0.469, compared to RELM, which only requires 0.123 of a second.

### 5.2. Scenario Using the REDD Dataset

REDD contains the active power readings of six homes at a frequency of 1/3 Hz. We use measurements from residence 1 for the REDD scenario. Within this household, a range of appliances is present, including an oven, refrigerator, dishwasher, kitchen oven, lighting, washer-dryer, microwave, bathroom Ground Fault Interrupters (GFI) outlet, heat pump, stoven, and an unidentified device. In this case, the system generates 2723 images illustrating the activities of household appliances.

In this scenario, the method produces the results displayed in Table 4, employing the REDD database from Kolter and Johnson [42]. As indicated by Table 4, to meet the threshold ψ = 0.99 imposed by the CEV, it is necessary to utilize 25 components, i.e., *q* = 25, for the REDD dataset. As mentioned earlier, GS and K-CV determine the most suitable hyperparameters for the models. For the ELM architecture, the hyperparameter search specified $\eta_{\left(\mathrm{optimal}\right)}$ = 100. In this scenario, the search found $\eta_{\left(\mathrm{optimal}\right)}$ = 400 and $\lambda_{\left(\mathrm{optimal}\right)}$ = 0.1 for the RELM architecture.

Table 5 presents a direct model performance comparison. The results reveal a significant difference in terms of accuracy when we compare the ELM and RELM models, this advantage is approximately 2.7% for RELM. Concerning F1, the difference between the ELM and RELM models is even more noteworthy. Here, the RELM architecture presents an advantage of 2.78% for the F_1_ metric. Ultimately, the RELM classifier achieves the highest agreement between predicted and expected values, with κ = 0.9388.

Table 6 provides a training time comparison of the employed ML models. In this scenario, the training time for the RELM is 0.123 of a second, followed by the ELM, with 0.045 of a second. Although, in this scenario, the ELM has the shortest training time, this difference is indeed minimal.

In conclusion, the results suggest that the RELM model is the preferred choice for load recognition in this dataset. When compared again to the second-best model in the literature, SVM, the superiority of RELM is evident across all metrics. SVM achieves 96.31%, 96.36%, and 0.9381 for accuracy, F_1_, and κ, respectively, while RELM reaches 96.53% for accuracy, 96.48% for F_1_, and 0.9388 for κ. Furthermore, RELM has the shortest training time. While SVM requires 0.167 s for training, RELM only needs 0.082 s.

### 5.3. State-of-the-Art Methods Comparison

Table 7 compares the proposed system with 11 state-of-the-art methods for load recognition. In this manner, Table 7 demonstrates that all the approaches may exhibit common structural characteristics, such as feature extraction procedures and earning models. However, solely the proposed system and the works of Cabral et al. [7] and Heo et al. [13] include event detection stages. Only the proposed system and the reference Cabral et al. [7] contain procedures for optimizing ML models.

Feature extraction procedures are vital structures for load recognition systems. However, the more characteristics we need to extract, the more computationally expensive the system becomes. The feature extraction procedures employed in the works of Qaisar and Alsharif [14], Zhiren et al. [16], Mian Qaisar and Alsharif [8], Soe and Belleudy [15], Faustine and Pereira [10], and Heo et al. [13] extract more than two types of characteristics from the signals, i.e., these methods need more information about the electrical signals to work. In this sense, the methods of Cabral et al. [7], Huang et al. [12], and our system use techniques to decrease the computational complexity. In the case of Cabral et al. [7] and Huang et al. [12], they use the PCA and in our case, we use the NCA technique. But comparing our system with the reference Cabral et al. [7], we require a smaller volume of data due to the utilization of a reduced number of components through NCA. While Cabral et al. [7] employ 269 components for REFIT and 35 for REDD, we require only 228 for REFIT and 25 for REDD.

The works of Faustine and Pereira [10], Matindife et al. [11], and De Baets et al. [9] use a more computationally complex architecture as a learning model, the CNN. However, as the last column of Table 7 shows, using more complex models does not guarantee a superior result. In this sense, the work of Matindife et al. [11] achieves 83.33% accuracy, while our method obtained 97.24% accuracy with RELM. Furthermore, only our study and the reference Cabral et al. [7] apply GS with K-CV to improve the performance of ML models. Nevertheless, the novelties in our work provide superior performance compared to Cabral et al. [7]. While reference Cabral et al. [7], the second-highest performing, shows 96.88% accuracy, using SVM for the REFIT dataset, we achieve 97.24% accuracy using RELM for the same dataset. It is relevant to point out that there is no consensus on the most suitable dataset for analyzing the methods. But in terms of metrics, most papers use accuracy as the principal evaluation metric, followed by F1-Score or a variation thereof, such as F_1_ in Cabral et al. [7]. Once again, our method shows the highest value for both metrics, 97.24% for accuracy and 97.14% for F_1_, while the reference Cabral et al. [7] shows 96.88% for accuracy and 96.61% for F_1_. Our method also has the highest values for the κ metric, demonstrating that our system has the highest rate of agreement for the results reached.

By examining Table 7, it is worth mentioning that the accuracy reported in studies can be affected by the metrics and datasets used. For this reason, reliable studies present more than one metric for performance analysis and use more than one database. Studies that focus only on accuracy are limited in terms of method reliability. Therefore, additional performance metrics, such as F_1_ and κ, are essential. Moreover, the dataset can influence the performance results. Studies that use more than one database tend to present a more robust analysis of the model’s performance.

## 6. Conclusions

This manuscript presents significant improvements in the area of load recognition. This work is the first to use NCA for enhanced feature extraction and RELM to classify household appliances. Furthermore, our study is also a pioneer in verifying NCA-ELM and NCA-RELM pairs in load recognition. When employing RELM, our analysis unveils an exceptionally short training time of less than 1 s for both databases, REFIT and REDD. Specifically, during the examination of training time, we attained a training duration of 0.082 s with RELM. This duration is shorter than that achieved with the SVM architecture in Cabral et al. [7], which was the state of the art up to the present, with a time of 0.167 s. By analyzing the accuracy metrics, F_1_ and κ, the superiority of RELM is evident. When compared to the state-of-the-art, RELM outperforms SVM in all the metrics. Whereas the SVM shows values of 96.88%, 96.61%, and 0.8375 for accuracy, F_1_, and κ in the REFIT database, RELM achieves values of 97.24%, 97.14%, and 0.8300 for accuracy, F_1_, and κ in the same database. The superiority of RELM extends to the REDD dataset, where SVM shows 96.31% accuracy, 96.36% F_1_, and 0.9381κ, whereas RELM reaches 96.53%, 96.48%, and 0.9388, respectively, for the same metrics. The proposed system demonstrates that the joint use of NCA and RELM is a viable and more robust alternative for load recognition, making the NCA-RELM pair a reliable and promising implementation.

The main drivers of the differences between the proposed method and competitors can be attributed to the innovative use of NCA with the RELM model. Feature extraction through NCA provides superior class separability and the application of the ELM yields higher reliability in identifying household appliances. This double focus is the main guide behind the improved performance of our system compared to existing approaches and paves the way for new possibilities for load recognition in HEMS systems. In this context, we suggest verifying additional datasets to evaluate their real-time implications for future research.

## Figures and Tables

**Figure 1 sensors-24-02274-f001:**
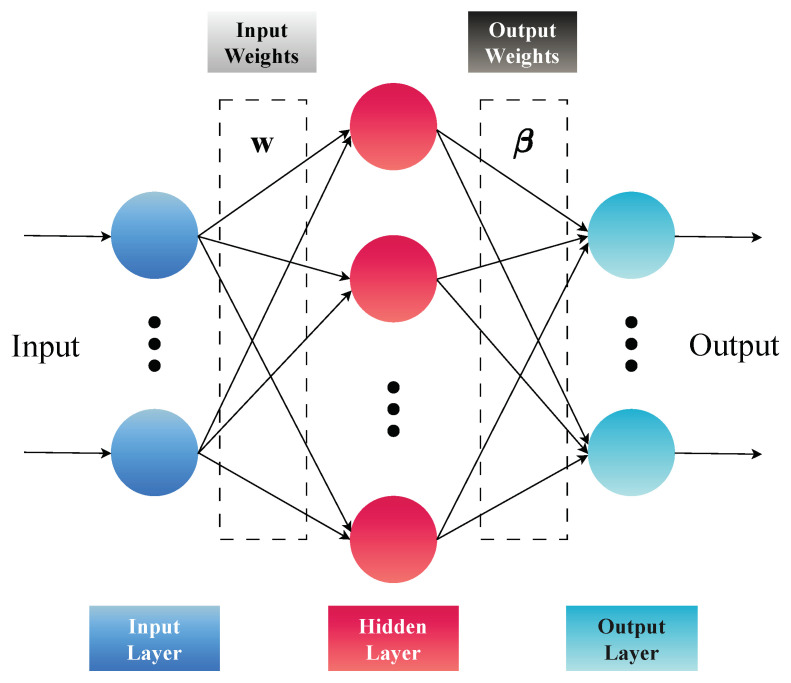
ELM-standard model (adapted from Zhao et al. [39]).

**Figure 2 sensors-24-02274-f002:**
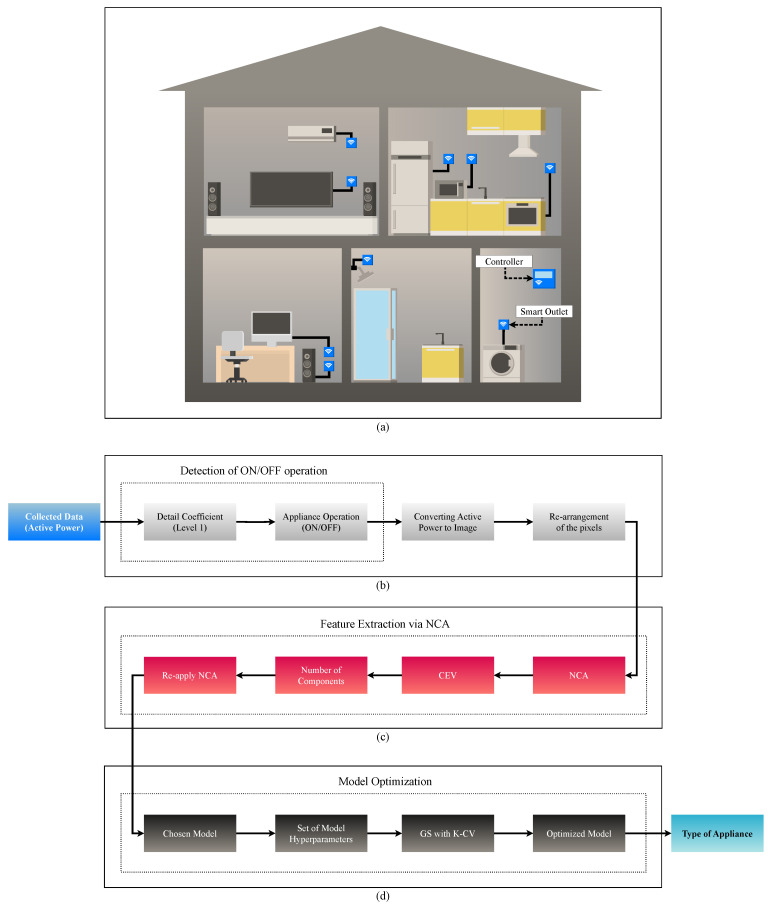
Comprehensive visualization of the Load Recognition System. This figure outlines the four stages of the process, starting with (**a**) the collection of active power through HEMS. The next phase (**b**) involves the detection of appliances’ ON/OFF status and preliminary data handling. In the sequel, (**c**) feature extraction is conducted using the NCA technique. The process culminates with (**d**) the optimization of ML models for improved classifier performance and the identification of the operational appliance type.

**Table 1 sensors-24-02274-t001:** Evolution of the CEV according to the increment in the number of components (Comp.) for the REFIT dataset.

Comp.	CEV	Comp.	CEV	Comp.	CEV	Comp.	CEV	Comp.	CEV
1	0.0862	53	0.8421	105	0.9330	157	0.9682	209	0.9858
5	0.2761	57	0.8539	109	0.9368	161	0.9700	213	0.9867
9	0.3983	61	0.8643	113	0.9404	165	0.9717	217	0.9876
13	0.4907	65	0.8736	117	0.9438	169	0.9733	221	0.9885
17	0.5669	69	0.8820	121	0.9469	173	0.9748	225	0.9894
21	0.6275	73	0.8896	125	0.9499	177	0.9763	227	0.9898
25	0.6792	77	0.8966	129	0.9527	181	0.9777	228	0.9900
29	0.7202	81	0.9031	133	0.9553	185	0.9790	229	0.9902
33	0.7505	85	0.9091	137	0.9578	189	0.9802	233	0.9910
37	0.7751	89	0.9147	141	0.9601	193	0.9814	237	0.9917
41	0.7961	93	0.9198	145	0.9623	197	0.9826	241	0.9924
45	0.8135	97	0.9245	149	0.9644	201	0.9837	245	0.9931
49	0.8287	101	0.9290	153	0.9664	205	0.9847	249	0.9938

**Table 2 sensors-24-02274-t002:** Performance of the classifiers for the REFIT Scenario.

Classifier	Accuracy	F_1_	𝜿
ELM	96.88%	96.59%	0.7910
RELM	97.24%	97.14%	0.8300

**Table 3 sensors-24-02274-t003:** Training time in seconds for the REFIT Scenario.

RELM	ELM
0.082	0.191

**Table 4 sensors-24-02274-t004:** Evolution of the CEV according to the increment in the number of components (Comp.) for the REDD dataset.

Comp.	CEV	Comp.	CEV	Comp.	CEV	Comp.	CEV	Comp.	CEV
1	0.8992	7	0.9631	13	0.9788	19	0.9860	25	0.9904
2	0.9223	8	0.9669	14	0.9803	20	0.9868	26	0.9909
3	0.9387	9	0.9704	15	0.9817	21	0.9876	27	0.9915
4	0.9474	10	0.9732	16	0.9829	22	0.9884	28	0.9920
5	0.9533	11	0.9754	17	0.9841	23	0.9891	29	0.9924
6	0.9586	12	0.9771	18	0.9851	24	0.9897	30	0.9929

**Table 5 sensors-24-02274-t005:** Performance of the classifiers for a REDD Scenario.

Classifier	Accuracy	F_1_	𝜿
ELM	93.82%	93.70%	0.8913
RELM	96.53%	96.48%	0.9388

**Table 6 sensors-24-02274-t006:** Training time in seconds for REDD Scenario.

ELM	RELM
0.045	0.123

**Table 7 sensors-24-02274-t007:** Comparison of state-of-the-art approaches.

Load Recognition Strategies	Event Detection Stage	Feature Extraction Stage	Learning Model	Model Optimization	Metrics	Best Result	Model for the Best Result	Dataset for the Best Result
Our System	DWT	NCA	ELM and RELM	GS with K-CV	Accuracy, F_1_, and κ	97.24% of Accuracy	RELM	REFIT
Ref. [7]	DWT	PCA	DT, *k*-NN, RF, and SVM	GS with K-CV	Accuracy, F_1_, and κ	96.88% of Accuracy	SVM	REFIT
Ref. [14]	None	Extraction of electrical operating patterns	*k*-NN and SVM	None	Accuracy	95.40% of Accuracy	SVM	ACS-F2
Ref. [16]	None	Extraction of electrical quantity	ELM, AdaBoost-ELM, and SVM	None	Accuracy	94.80% of Accuracy	AdaBoost-ELM	Private
Ref. [8]	None	Extraction of energy consumption patterns from appliances	ANN and *k*-NN	None	Accuracy	94.40% of Accuracy	ANN	ACS-F2
Ref. [15]	None	Extraction of electrical operating patterns	CART, *k*-NN, LDA, LR, NB, and SVM	None	Accuracy	94.05% of Accuracy	*k*-NN	ACS-F1
Ref. [10]	None	Extraction of high-frequency features	CNN	None	F_1_-eb and F_1_-macro	94.00% of F_1_-macro	CNN	PLAID
Ref. [13]	RMS Threshold	APF	HT-LSTM	None	Accuracy and F_1_-Score	90.04% of Accuracy	HT-LSTM	PLAID
Ref. [19]	None	Stockwell transform	VPC	None	Identification percentage	90.00% of Accuracy	VPC	Private
Ref. [11]	None	GADF	CNN	None	Accuracy, precision, recall, F_1_-Score, and κ	83.33% of Accuracy	CNN	Private
Ref. [9]	None	VI trajectories	CNN	None	F_1_-macro, precision, and recall	77.60% of F_1_-macro	CNN	PLAID
Ref. [12]	None	PCA	LSTM-BP	None	F_1_-Score	45.49% of $F_{1}$-Score	LSTM-BP	REDD

## Data Availability

The access to the data underlying the findings of this study is not available due to privacy considerations and in accordance with company operational policies.

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
