# Peer review of "Load Recognition in Home Energy Management Systems Based on Neighborhood Components Analysis and Regularized Extreme Learning Machine"

_sensors, 2024, doi:10.3390/s24072274_

Round 1

Reviewer 1 Report

Comments and Suggestions for Authors

The authors have presented a model based on Neighborhood Components Analysis (NCA), extreme learning machine and regularized extreme learning machine for feature extraction and load recognition. The work is novel, interesting and its title reflects the proposed model correctly, however, it should be revised as per the following comments.  

·         The introduction section lacks the flow of the information, it should build the case for importance and practical applications of load recognition for justification of the proposed work.

·         The contribution section repeats the same thing many times. The mentioned six contributions are three which should be summarized accordingly. The contribution 6 is not a contribution, it is just a generic application. The sixth contribution should be removed, and its sentence should be grammatically correct.

·         The background section should mention the shortcomings of the existing work.

·         Figure 1 depicts the ELM-standard model, it should be cited properly.

·         Various typos are found in the manuscript. e.g. Line 201, “diferent”, Line 207, “Neverthenless”, Line 212, “sistem” etc. The whole manuscript should be checked for typos and grammatical errors.

Comments on the Quality of English Language

   Various typos are found in the manuscript. e.g. Line 201, “diferent”, Line 207, “Neverthenless”, Line 212, “sistem” etc. The whole manuscript should be checked for typos and grammatical errors.

Reviewer 2 Report

Comments and Suggestions for Authors

1. The abstract could be improved for clarity and conciseness with reference to Identifying major energy-consuming appliances in residential building components that needs to be considered for optimal system performance and better energy management systems.

2. How the Neighborhood Components Analysis is done. What are the different component that is considered for the analysis.

3. In line number 220, If hardware details are not the focus, then it is not required to mention the specification as "390 MHz processor" and "512MB of RAM". It's unclear why the specific hardware specifications are relevant to the discussion. If the manuscript is focusing on Home Energy Management System (HEMS) and its functionality, then its hardware specifications should be mentioned and it should be explained clearly.

4. Mentioning a "390 MHz processor" and "512MB of RAM" without specifying the exact model or type of processor and RAM doesn't provide a clear understanding of the hardware capabilities. For technical evaluations or comparisons of Load Recognition, it's important to provide detailed specifications.

5. What is the reference used to frame the data available in Table:1 & Table:4

6. The primary focus on conclusion yield promising results, and certain limitations could be focused, for further validation across various datasets and real-time implication.

Reviewer 3 Report

Comments and Suggestions for Authors

This paper introduces a novel approach for load recognition, combining Neighborhood Components Analysis (NCA) technique and Regularized Extreme Learning Machine (RELM) models. The authors showcase the efficiency of their approach by comparing the results from its application to two datasets against state-of-the-art alternatives, in terms of key accuracy indicators. The topic of the paper is interesting and fits the journal, displaying a high level of innovation.

However, before considering the paper for publication in MDPI’s Sensors journal, I recommend that the authors address the following points, or adequately justify their choices should these suggestions not be implemented. My feedback is mainly related to better explaining the motivation behind each methodological component, explaining the drivers behind the reported differences between the results of the proposed methodology and state-of-the-art methods, and subsequently the significance of these differences. Due to the nature of these comments, I recommend a minor revision for the paper so that the authors improve their paper from the suggested perspectives. The proposed revisions are detailed below.

- The contributions of the paper in the abstract could be described as part of a cohesive narrative instead of listing them in bullet points. I propose the following structure for the abstract: Background, Research Gap, Methodology Description, and Key Results/Conclusions. Refrain from using phrases like ‘’In conclusion’’ in the abstract.

- The keyword selection appears overly broad, while many keywords are repeated in the title of the paper. Refine the list to include the most pertinent keywords not already mentioned in the title.

- In-text citations should follow the format ‘’Author name et al. [X]’’. This comment applies to many parts of the text (e.g., ‘’Heo et al. use Amplitude-Phase-Frequency (APF) [13].’’).

- Section 1.1. seems redundant given the preceding section's summary of contributions. Methodological improvements should be discussed also from the perspective of why they are important and how they fill the gap in the literature. For example, why using NCA for feature extraction in the load recognition problem is important and what gaps in the literature does this action address? Point 4: Try to give a context of the reported accuracy, e.g., how much the proposed method overperforms its counterparts? Do the same for point 5. Finally, is point 6 really a contribution? Isn’t something that one would be aware without reading this section?

- Use of poor language at the last paragraph of section 1. It must be revised from that regard (e.g., ‘’introduce the Background’’)

- In Section 2.1, key technical terms like active and reactive power should be described upon their first appearance in the text, to broaden the audience that the paper refers to.

- Line 137: ‘’we use Wavelet 137 transform to detect the ON/OFF status of the appliances according to references’’ Justify the selection

- I propose avoiding so strong expressions ‘’The undeniable advantage of some feature extraction techniques…….’’

- Both in figures and tables, an abbreviation must be defined at first use (see for example ELM in Figure 1). For Figure 2, I propose describing in its title and in the text first the whole figure composed of all four panels, in turn delving into each of its panels.

- For the boxes in the text describing algorithms (e.g., Feature Extraction via NCA), I recommend using more descriptive titles. Also, I propose transferring these boxes in the Appendix as they don’t offer a lot in the main body of the paper.

- I propose using a more descriptive title for Section 4. You need also to explain the rationale behind selecting the three accuracy metrics used in this study.

 - Line 318: autonomously à endogenously

- ‘’we use a threshold of ψ = 0.99… use K = 10 folds…… a step size of 100…’’ Why did you make these assumptions? Was the sensitivity of results against the variability of these parameters tested?

- In Table 2, can the differences between the proposed classifier and ELM be considered significant? That said, if we compare again the two classifiers by amending some of the input assumptions or dataset, would the proposed classifier outperform ELM?

- ‘’The present work provides significant progress in the load recognition issue. For the first time in the literature’’ How do you define significant? I propose avoiding strong language without sufficient justification. Rather, I highly recommend explaining the drivers behind observed differences instead of just reporting numbers.

- In Table 7, is the reported accuracy across studies affected by the utilized metrics and datasets. These are critical aspects that the authors should touch on, integrating these insights in study's limitations and suggestions for future research.

-Conclusions: try to derive deeper conclusions instead of just reporting numbers. What are the key drivers behind the differences between the proposed method and state-of-the-art methods? Also, the authors should expand conclusions by describing suggestions for future research in the field. Future research should also be connected with the limitations of this study that the authors must explicitly describe. For example, how would the proposed future research assist in addressing the limitations of the study?

Comments on the Quality of English Language

The paper is generally well-written. However, it could benefit from a refinement of its language at some points (I provide reccomendations to the authors for this on my review report).

Round 2

Reviewer 2 Report

Comments and Suggestions for Authors

The author addressed all comments raised by the reviewer and revised the manuscript based on reviewer comments and hence it can be progressed further as this as this work is useful for the researchers who works under energy management domain. This work makes a significant contribution towards energy management domain by providing valuable insights particularly in incorporating Neighborhood Components Analysis (NCA) and Regularized Extreme Learning Machine (RELM) techniques, which significantly improve load recognition accuracy in Home Energy Management Systems (HEMS) and practical implementation of advanced machine learning algorithms for load recognition tasks.